# Satellite-Based Flood Mapping through Bayesian Inference from a Sentinel-1 SAR Datacube

Bernhard Bauer-Marschallinger [1,*], Senmao Cao [2], Mark Edwin Tupas [1,3], Florian Roth [1], Claudio Navacchi [1], Thomas Melzer [1], Vahid Freeman [4] and Wolfgang Wagner [1,2]

1   Remote Sensing Research Group, Department of Geodesy and Geoinformation, TU Wien, 1040 Vienna, Austria; mark.tupas@geo.tuwien.ac.at (M.E.T.); florian.roth@geo.tuwien.ac.at (F.R.); claudio.navacchi@geo.tuwien.ac.at (C.N.); thomas.melzer@geo.tuwien.ac.at (T.M.); wolfgang.wagner@geo.tuwien.ac.at (W.W.)
2   EODC Earth Observation Data Centre for Water Resources Monitoring, 1030 Vienna, Austria; senmao.cao@eodc.eu
3   Department of Geodetic Engineering, University of the Philippines Diliman, Quezon City 1101, Philippines
4   Space Program, Spire Global, 2763 Luxembourg, Luxembourg; vahid.freeman@spire.com
*   Correspondence: bbm@geo.tuwien.ac.at

**Abstract:** Spaceborne Synthetic Aperture Radar (SAR) are well-established systems for flood mapping, thanks to their high sensitivity towards water surfaces and their independence from daylight and cloud cover. Particularly able is the 2014-launched Copernicus Sentinel-1 C-band SAR mission, with its systematic monitoring schedule featuring global land coverage in a short revisit time and a 20 m ground resolution. Yet, variable environment conditions, low-contrasting land cover, and complex terrain pose major challenges to fully automated flood monitoring. To overcome these issues, and aiming for a robust classification, we formulate a datacube-based flood mapping algorithm that exploits the Sentinel-1 orbit repetition and a priori generated probability parameters for flood and non-flood conditions. A globally applicable flood signature is obtained from manually collected wind- and frost-free images. Through harmonic analysis of each pixel's full time series, we derive a local seasonal non-flood signal comprising the expected backscatter values for each day-of-year. From those predefined probability distributions, we classify incoming Sentinel-1 images by simple Bayes inference, which is computationally slim and hence suitable for near-real-time operations, and also yields uncertainty values. The datacube-based masking of no-sensitivity resulting from impeding land cover and ill-posed SAR configuration enhances the classification robustness. We employed the algorithm on a 6-year Sentinel-1 datacube over Greece, where a major flood hit the region of Thessaly in 2018. In-depth analysis of model parameters and sensitivity, and the evaluation against microwave and optical reference flood maps, suggest excellent flood mapping skill, and very satisfying classification metrics with about 96% overall accuracy and only few false positives. The presented algorithm is part of the ensemble flood mapping product of the Global Flood Monitoring (GFM) component of the Copernicus Emergency Management Service (CEMS).

**Keywords:** Sentinel-1; SAR; flood mapping; automatic flood monitoring; time series analysis; Bayes inference; datacube

## 1. Introduction

Floods are the most frequent and second-most costliest natural disasters worldwide, making up a share of 43% of the recorded disaster events in 1998–2017 and affecting over 2 billion people, according to the survey of the United Nations Office for Disaster Risk Reduction (UNISDR) [1]. For the period 2008–2018, the International Federation of Red Cross and Read Crescent Societies (IFRC) counted 730 million flood-affected people, representing 52% of world's disaster victims that suffered 23% of economic damage and 7% of recorded deaths related to natural disasters over that period [2]. With climate change and

an intensifying water cycle, human losses, infrastructural damages, and economical losses are expected to increase in the future [3], a trend that will be exacerbated by a growing flood vulnerability due to urbanisation, population growth, and land cover change.

In light of this threat, knowing where and when floods appear is imperative to authorities and disaster management units around the globe. Any effective flood management requires timely and detailed flood maps to enable preparation, planning, and rapid response to the local flood risk. Here, satellite remote sensing offers a rich data source [4], allowing to assess the situation from the bird's eye perspective. A recent review article [5] offered a synopsis on satellite remote sensing in flood management, expounding on the technological developments since the 1970s and on how the European Copernicus program with their Sentinel satellite fleet can assist in various phases of flood management. Likewise, rainfall observations from space are well-established, and are used for example in riverine flood modelling, which shows significant potential in flood forecasting [6].

As to **flood extent mapping**, state-of-the-art imaging sensors revolve along low Earth orbits and scan our planet at a resolution of some 10 m and achieve up to a daily measurement frequency, depending on the geographic location, satellite mission constellation, and employed sensor technology. While optical sensors allow an easy-to-interpret analysis, they are often blocked by cloud cover during flood situations when severe weather conditions are predominant [5,7]. In contrast, Synthetic Aperture Radar (SAR) sensors have day-and-night and all-weather capabilities [8], and can observe the situation at all overpasses. Although the peak extent of a flood might not coincide with a satellite overspass and is usually not recorded, SAR systems offer observations more regularly and can capture the flood's rise and fall. In conjunction with their high sensitivity to water occurrences, satellite-based SARs are excellent instruments to directly map flooded surfaces on a regional scale, and are well suited for automatic and global flood monitoring. That said, the employed scientific algorithms need to account for land cover and environmental effects on the radar signal. Due to the observational distance from space and the resolution limits of SAR sensors, spatial detail in flood delineation can miss the requirements of users who work on the local scale or in areas with high complexity from topography, vegetation, or buildings. To enhance delineation fidelity, SAR data can be combined with elevation or optical datasets [9], or be assimilated into hydraulic models [10,11].

Methods that **map floods directly** in SAR imagery are well established and allow for fast and straightforward analysis on an image/scene level, as water bodies typically feature a strong contrast to other surfaces in SAR images. The reason lies within the difference between microwave scattering mechanism over water and land surfaces, and the side-looking geometry of SAR systems. A specular reflection of the radar pulses by the water surfaces leads to backscatter intensities received at the sensor that are much lower than for most other land cover types [12]. This physical mechanism renders the mapping of open, calm water in principle rather straightforward. Accordingly, many SAR-based flood mapping algorithms have been developed based on image histogram analysis, leading to thresholds that divide the area into low and high backscatter regions and, respectively, into water and non-water.

Even when applied to only **single SAR images**, thresholding often produces quite satisfying results (e.g., [13–15]), and shows robust performance when refined with automated hierarchical thresholding in image-tiles, fuzzy logic post classification, and region growing (e.g., [16,17]). However, this requires a suitable reference map on permanent and seasonal water bodies to distinguish flooded areas. Furthermore, applying such algorithms over larger regions in an automated fashion is challenging due to the complexity of the terrain, heterogeneous land cover, and varying environmental conditions. In particular, maps based on single images often contain false positives over land surfaces with backscatter signatures as low as that from water surfaces, i.e., an overestimation due to non-flood pixels detected as flooded pixels. Such *water look-alikes* are typically found in areas featuring smooth surfaces (e.g., tarmacs, sands, salt pans), dry and sparse vegetation (e.g., prairie

grasslands), bare rock grounds, and in areas located in the image's radar shadow that appears along high mountain ranges, forest lines, or buildings.

The impact of these confounding effects can be minimised by using **change detection** approaches, which (1) are less sensitive to the generation of false positives and (2) directly yield flood areas instead of water areas. Here, changes between two subsequent measurements are attributed to sudden changes occurring on the ground, transforming the flood mapping issue to a classification problem between change and no-change. Following the computation of a difference image (i.e., the change image), different histogram threshold approaches (e.g., [10,18]) can be applied to generate the binary classification. Such change detection methods assume that one type of change (i.e., a decrease of backscatter due to the specular reflection on water bodies) dominates all other changes, and therefore, might produce misclassifications of non-water pixels due to low backscatter from dry soil conditions. Yet, the study of [19] showed that such false positives can be effectively reduced with dual-image processing approaches. Further enhancements are achieved again with method refinements, such as hierarchical image tiling and region growing, e.g., recently in [20].

Today, we have access to a large number of SAR satellites from multiple past and current missions, including Envisat ASAR, Radarsat, TerraSAR, and Sentinel-1. This allows us to go one step further and use the availability of many observations distributed over time. Then, we can build **SAR backscatter time series** and map floods based on the signal's deviation from a priori statistical parameters. For example, in the method of [21], the rationale is that usually, land pixels can be classified as flooded when they show a distinct deviation from the expected seasonal backscatter that is modelled with harmonic functions.

Another established approach to address ambiguities in SAR images between flooded and non-flooded is to produce a measure of **uncertainty** from the classification process. In this regard, probabilistic methods, specifically Bayesian inference and its multi-node extensions—so-called Bayesian networks—are popular choices [15,22–25]. Here, the probability of a given SAR backscatter pixel is assessed against predefined flooded and non-flooded probability distributions, and the flood mapping is realised through selection of the more probable class, along with a certainty value. The required distributions can be inferred from historic observations and—when ingesting local time series—may establish pixel-specific parametrisation.

The employment of time series for the detection of floods in satellite images necessitates in practice the formation of a **datacube**, where historic and new images are unified through well-defined methods on SAR preprocessing, gridding, and file storage. In a datacube, the temporal and spatial dimensions are treated alike, and therefore, each SAR image eligible to flood mapping can be directly compared with the entire backscatter history, allowing to implement on a per-pixel basis different sorts of change detection algorithms in a straightforward and efficient manner. Subsequently, pixel-specific thresholds and model parameters can be applied without the need for spatial re-gridding or resampling and allow fast flood classification. Recently, ref. [26] developed on the Google Earth Engine platform a flood algorithm that combines processing with access to satellite datacubes [27], using a decision tree informed from antecedent Sentinel-1 and Landsat time series.

Effectively, a datacube enables (1) a more robust handling of land surface heterogeneity, (2) the a priori determination of regions where open water cannot be detected for physical reasons (e.g., dense vegetation, urban areas, deserts), (3) the estimation of the flood mapping's uncertainties, and (4) the generation of historic water extent maps, essentially as a by-product of the model calibration, which may serve as a reference for distinguishing between floods and the normal seasonal water extent. For instance, ref. [28] showed that time series analyses applied on SAR data archives are also well suited to improve the characterisation of permanent water bodies, and [29] derived an exclusion layer to remove overestimations of flood extent in arid regions.

In this paper, we present in Section 2 our new method based on a Sentinel-1 datacube, followed by a description of the input data collection, the flood mapping algorithm with its statistical model, and its procedures to identify insensitivities. For a major flood event in

Greece/Thessaly in 2018, Section 3 interprets the generated parameters and discusses an in-depth investigation of the method's performance, and Section 4 draws conclusions and considers future research directions.

## 2. Data and Methods

We take up the datacube approach for flood mapping, and present a time series-based detection method for Sentinel-1 radar data, using a simple Bayes classifier in conjunction with data-driven masks for low-sensitivity areas.

### 2.1. Our New Method Based on a Sentinel-1 Datacube

The 2014-launched Sentinel-1 mission [30] of the European Earth observation programme Copernicus employs C-band SAR instruments (CSAR) operated at a 5.5 cm wavelength. It is the first SAR mission that is dedicated to systematic backscatter acquisitions, with a two-satellite-constellation scanning all global land masses at 10 m sampling within 12 days (Note: Since December 2021, Sentinel-1B suffers from an operational anomaly and its CSAR sensor is not active, reducing global coverage roughly by a factor of 2. See also Section 4). With this, the mission offers an unprecedented spatio-temporal coverage as well as radiometric accuracy and stability, and fuels many applications through enabling the retrieval of geophysical variables as, e.g., soil moisture [31,32], vegetation density [33,34], crop status [35,36], or snow depth [37]. However, Sentinel-1's sensor design and acquisition strategy pose new challenges, as they constitute also a break with former C-band SAR missions ERS-1/2, Radarsat-1/2, and Envisat ASAR, as (1) it provides VV-polarised radar observations over land areas and (2) its satellites follow a strict acquisition scenario, scanning the ground under repeating viewing angles, and thus, limiting the range of observations angles. While the VV-channel is considered most suitable for the detection of water surfaces through its generally higher sensitivity on this matter [17,38], the limited number of observations angles poses a challenge for backscatter normalisation, which is usually required to obtain consistent classification results within the stretched image extent [39]. Moreover, the stationary orbit configuration of Sentinel-1 generates a discriminative swath footprint pattern (as, e.g., discussed in [32])—with some areas observed only by one or two orbits, and many among them with a narrow or even non-existent incidence angle range—and as a consequence, the incidence angle normalisation suffers from high uncertainty or relies on spatial proxies [40]. Some studies on water mapping ignore the incidence angle effect by arguing that Sentinel-1's incidence angle range is rather narrow [41], while others only use acquisitions from identical relative orbits, tolerating lower revisit frequency and less reliable model parameters.

Our here-presented change detection algorithm pursues a new strategy, exploiting the availability of historical backscatter measurements within an spatially extensive multiyear Sentinel-1 datacube. After reaching the (obvious) definition that flood is water where normally no water is, our three central statements are the following:

1.  For **all water bodies** around the globe, we assume that they have an identical C-band SAR backscatter signature, independent from local conditions such as depth, underwater ground, or turbidity. This assumption is justified by the fact that the penetration depth of microwaves into water is just a few millimetres at best. Under the conditions that the water bodies are open (i.e., not covered by vegetation), calm (i.e., not roughened by wind), and non-frozen, the backscatter signal is of universal character and primary related to the incidence angle. This allows us estimating global backscatter parameters for water bodies that particularly include flood bodies (hereafter *water distribution*), derived from selected Sentinel-1 measurements collected from calm and open water bodies. Eventually, we form these a priori backscatter distributions for a set of fine bins within the Sentinel-1 incidence angle range.

2.  Contrary, the backscatter signal over land is diverse and heterogeneous, and thus, we localise its parametrisation and retrieve the a priori local backscatter distribution for **each individual pixel over the landmasses**. Building upon the already available

Sentinel-1 datacube that comprises data since 2014, we estimate the local backscatter distribution per relative orbit geometry (hereafter *local distribution*) and do not apply any incidence angle normalization. We further assume that flooded conditions are highly infrequent, and hence, neglect their impact on the multiyear statistics, and declare the local distribution to represent non-flood conditions.

3.  For the actual **flood mapping**, the values of the incoming Sentinel-1 image are analyzed pixel-wise against the water distribution (respective to the incidence angle) and the local distribution (respective to the orbit). By means of *Bayesian inference*, it is then possible to derive the value's posterior probabilities of belonging to the water and the local distribution, and hence, to decide if the image pixel belongs to either the flood or the non-flood class. Applying the Bayes decision rule yields not only the class allocation, but also implicitly provides a probabilistic uncertainty measure at each pixel.

Accordingly, the algorithm fully exploits the entire Sentinel-1 signal history within the datacube, realised by a set of a priori computed statistical parameters that provide via a harmonic seasonality model a specific SAR characterisation of the Earth's land surface at the pixel level. Water surfaces are modelled globally and with respect to Sentinel-1's incidence angle dependency. With those parameters as input, and with the mathematical legacy of Bayes, the flood delineation procedure can be designed computationally relatively slim, it does not require any human interaction (e.g., on selection of reference images), and it is hence most suitable for automatic global operations in *near-real-time (NRT)*. Limiting our water definitions to calm and open waters is a concession made to achieve our operational objectives (including automatisation), recognizing the various complications with the SAR modeling over roughened and overgrown waters. Such situations can be highly dynamic and require thematic flagging a posteriori, which is beyond this paper's scope.

However, the two obtained distribution parameter sets allow an a priori identification of our flood algorithm's no-sensitivity areas, where the water and local distribution are too strongly overlapping and the flood decision is not reliable or possible. This includes permanent and seasonal water bodies, as well as permanent low-backscatter pixels from water look-alikes, such as airports and motorways. Finally, a topography mask is applied to elevated areas where floods hardly occur, and hence, support the robustness of a potential automatic global service for detecting floods.

Figure 1 summarises the general workflow between the main components of our flood mapping method.

### 2.2. Sentinel-1 Datacube Formation

Observational input to our flood algorithm is generated by the C-band sensor (CSAR) onboard the Sentinel-1A and -1B satellites, operated in the Interferometric Wide-swath (IW) mode that is the mission's main operational mode over land and measures backscatter in dual polarisation (VV and VH). In IW mode, Sentinel-1 offers a systematic and regular revisit of 9 to 1 local observations within 12 days, following the mission's orbit cycle and its observation scenario (see https://sentinel.esa.int/web/sentinel/missions/sentinel-1/observation-scenario (accessed on 25 July 2022), details discussed in [32,40]).

For this study, we collected for the period 2015–2020 the VV-polarised IW mode Ground Range-Detected at High resolution (IWGRDH) products that hold backscatter amplitude data and are characterised by a 10 m pixel spacing, a nominal spatial resolution of 20 m $\times$ 22 m, and a radiometric accuracy of 1dB ($3\sigma$) [30].

The build of our Sentinel-1 IW datacube is detailed in our recent dedicated publication in [42] (together with a description of access options). As a brief summary here, all files underwent parallelised preprocessing comprising (1) precise orbit data usage, (2) image border noise removal (following an algorithm developed specifically for S-1 [43]), (3) thermal noise removal, (4) radiometric calibration, (5) terrain correction, yielding an intermediate image at 10 m pixel sampling in geographical coordinates, (6) reprojection onto the Equi7Grid, (7) downsampling with gdalwarp/cubicsplines to a 20 m pixel-size,

and (8) splitting into 300 km-sized tiles; (gdalwarp accessed on 25 July 2022 at https://gdal.org/programs/gdalwarp.html).

**Flood mapping algorithm** based on parameters from Sentinel-1 datacube

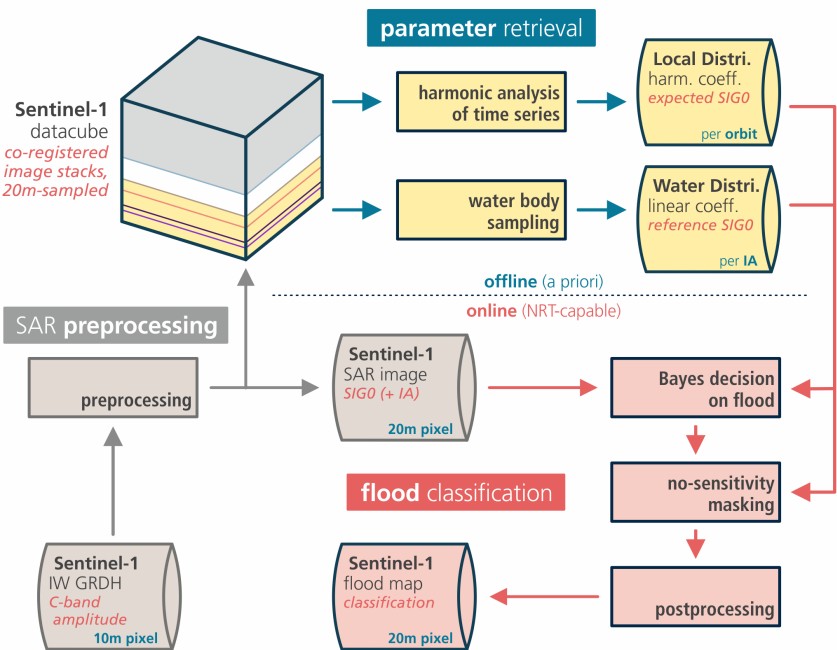

**Figure 1.** Schematic overview of the flood mapping algorithm's main components and data flow. Gray module: SAR preprocessing (not subject of this publication); blue module: offline/precomputed parameter retrieval; red module: the online/NRT flood classification. SIG0: sigma nought backscatter coefficient ($\sigma^0$). IA: incidence angle ($\theta$).

The Equi7Grid [44] is a global spatial reference system designed to handle efficiently the archiving, processing, and analysis of high resolution raster data over land, as it minimises data oversampling and preserves geometric accuracy. Its features have been found most beneficial in global terrain analysis by [45], and its design allows spatial accuracy in flood mapping around the globe.

The choice of the 20 m pixel sampling (instead of 10 m) is motivated by noise reduction. A Sentinel-1 image, owing to the nature of the SAR observation technique, inevitably carries speckle and signal noise, and consequently, the effective resolution is somewhat coarser than the nominal resolution. Although the processing to GRDH already dampens the noise level, it can be effectively reduced further through spatio-temporal averaging and filtering, closing the gap between nominal and actual resolution. For the flood mapping with Sentinel-1 IW images, we considered a downsampling to 20 m as a good compromise between noise reduction and resolution power for water body delineation. Moreover, the data size (reduced by factor ∼4) significantly reduced the required storage and processing power, speeding up the parameter generation and flood estimation. Details on the used SAR preprocessing methods and how they are employed in High-Performance Computing (HPC) environments can be found in the studies of [40,46,47].

The obtained SAR images hold $\sigma^0$ *(sigma nought) backscatter coefficient* values in decibel (dB). They are co-registered and time-stacked over the Equi7Grid-tile EU020M_E054N006T3 (covering our study site in Greece, cf. Section 2.7), ranging from January 2015 to December 2020, and providing ∼600 individual measurements from orbits D080 (descending orbit direction) and A175 (ascending) for the study area centre. The other five local orbits had no overpass during the flood event. These images build, together with information on the topography (based on digital elevation models, DEMs), the basis for the parameter generation, flood mapping, and masking presented in this study. We note that the European Space Agency (ESA), as the primary provider, slices Sentinel-1 the IW products along-track

per 25 s sensing time (equivalent to about 170 km in azimuth direction), and we forward the initial slicing and timestamps to our preprocessed datacube. In the course of the flood mapping, due to the algorithm's design aiming for NRT operations, adjacent Sentinel-1 slices stemming from the same overpass are not spatially merged, and a thin line along the product slicing may remain unclassified.

As to the observation geometry, the projected local incidence angle (PLIA) values are available as a by-product of the terrain correction step of the SAR preprocessing chain. For the purpose of flood mapping, PLIA describes appropriately the radar geometry over flat terrain and acts as the *incidence angle (IA, θ)* in this study. Because of the self-repeating orbit geometries of the Sentinel-1 mission—the satellite positions are maintained within an orbital tube of 50 m ($1\sigma$) [30]—almost identical observation angles are established at each overpass. Globally, the Sentinel-1 mission has 175 (repeating) relative orbits, with locally up to 9 orbits. Consequently, when working with Sentinel-1 data separately per relative orbit (indexed in the following by $\rho$), a pixel's value for $\theta_\rho$ can be assumed constant. We capitalise on this and use as input to our algorithm a set of constant $\theta_\rho$ values, which are computed a priori and per-orbit as average $\theta$ of the Sentinel-1A+B observations of the year 2020.

Our notation in this paper uses subscripts for data-based parameters, which are estimated a priori and stored on disk, e.g., $\theta_\rho$, whereas variables appearing in runtime are notated by parentheses, e.g., $\sigma^0(t)$.

### 2.3. Backscatter Parameters for Water and Land Surfaces

Following our initial three central statements on our approach to map floods, where an SAR backscatter image shows a water signature instead of the expected local (land) signature, we generate dedicated statistical parameters from the Sentinel-1 multiyear data archive. We use these parameters to compute the posterior probabilities for the classes *flood* and *non-flood*, which are subsequently input to the Bayes decision rule (Section 2.4).

Based on the premise that flood bodies show the same signature as regular water bodies, we infer the *flood backscatter probability distributions* from the manually collected water distribution (Section 2.3.1).

To obtain the *non-flood backscatter probability distributions*, we derive from the pixel's time series the so-called *harmonic coefficients* to model the local seasonal signal holding the expected values within the yearly cycle. With this, we declare the expected local distribution to represent the non-flooded conditions, irrespective of the actual land cover including permanent or seasonal water bodies—as we define floods as water occurring where it normally does not (Section 2.3.2).

#### 2.3.1. Flood Backscatter Probability Distributions

In radar images, water surfaces show typically a strong contrast to land surfaces. This was recently confirmed for Sentinel-1 CSAR and permanent water bodies in the global study of [40]. Consequently—and important to flood mapping—temporarily inundated surfaces introduce a drop in the backscatter time series of an affected land area. To differentiate these changes from other effects with similar outcome, a detailed knowledge of the backscatter behaviour over water surfaces is required.

As demonstrated in [25], the SAR backscatter behaviour over water surfaces can be represented by a normal distribution, and it can be retrieved from a representative collection of SAR measurements over water bodies. Following this approach, we collected various backscatter observations $\sigma_w^0(\theta_w)$ along with the respective incidence angles $\theta_w$ over ocean and inland water surfaces from the Sentinel-1 datacube. Due to the typical increase of backscatter over water during wind or frost conditions, and the much reduced separability against land in such cases, the collection was thoroughly filtered for calm conditions based on visual image inspection. As in [25], we extracted the actual water surfaces by the use of global land cover data, and additionally removed the pixels on the edge line of each water body to avoid the influence of mixed land-water pixels. The

representative water collection was then aggregated through averaging per month and orbit, and comprises ∼1000 individual composites that cover the 2015–2016 period and 12 European Equi7Grid-tiles.

Before one can estimate water backscatter signatures, the strong linear relation between backscatter and incidence angle must be accounted for. To eliminate the impact of the incidence angle on the observed backscatter value, ref. [25] normalised ENVISAT ASAR backscatter to a reference incidence angle, while [15] applied an approach based on the assignment of reference angles to discrete classes (bins) and subsequent distribution sampling. Sentinel-1 CSAR, however, provides only a limited number of incidence angles per pixel, and with only one or two incidence angles per pixel, the underdetermined equation system would introduce high uncertainty. Therefore, our method developed for Sentinel-1 provides an equivalent approach without the need for normalisation.

With the backscatter probability density function (PDF) $p(\sigma^0 | F, \theta)$, we model the flood likelihood for any given incidence angle $\theta$ (generalising from $\theta_w$). Assuming specifically a conditional normal distribution, its PDF $n(\sigma^0 | \mu_w(\theta), s_w^2)$ is determined by its mean $\mu_w(\theta)$ and standard deviation $s_w$. If the relationship between $\mu_w$ and $\theta$ is linear and $s_w$ is constant, these parameters can be obtained by linear least squares. In order to verify our assumptions, within our water-backscatter collection $\sigma_w^0(\theta_w)$, we first sorted the backscatter samples along the incidence angles $\theta_w$ and grouped them in 0.5° bins, using two-sided rounding to the bins' centre values, noting that this binning size is considered precise enough to cover its impact on backscatter (Figure 2a).

**a)** Sentinel-1 **backscatter** over **water surfaces** against **incidence angle**

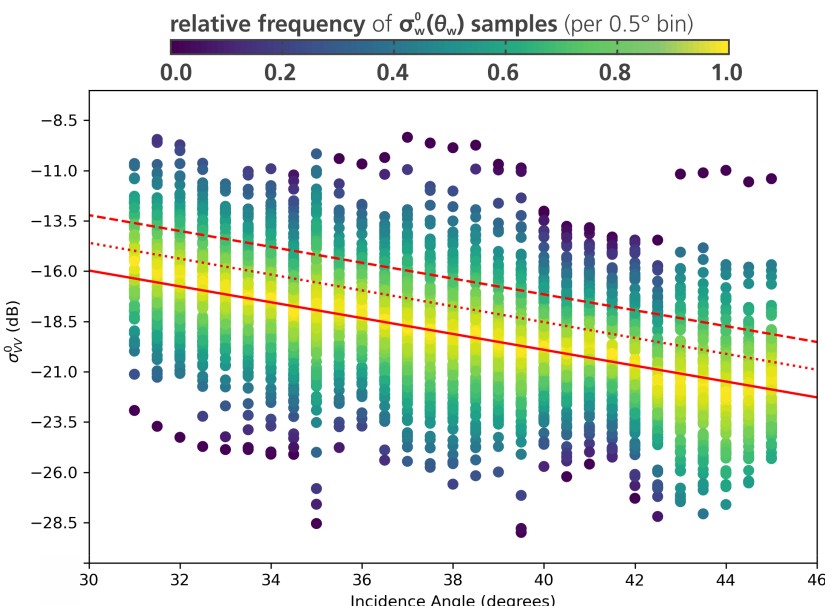

**b) Standard deviation** of $\sigma_w^0(\theta_w)$ within 0.5° bins

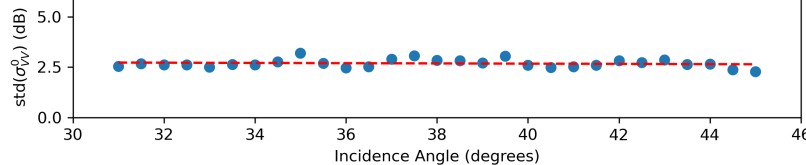

**Figure 2.** (**a**) Scatterplot of collected Sentinel-1 backscatter coefficients $\sigma_w^0(\theta_w)$ over water surfaces against incidence angles $\theta_w$, arranged within 0.5° incidence angle bins. The solid red line is the fitted linear function, while the dashed red line indicates one standard deviation, and the dotted line half of the standard deviation (used for masking in Section 2.5.2). (**b**) Standard deviation of backscatter coefficients within the respective incidence angle bins.

Obviously, the mean water-backscatter values within each bin show—as expected—a linear relation with the incidence angles, which can be parametrised as:

$$\mu_w(\theta) = \beta_1 \theta + \beta_0 \quad [\text{dB}] \tag{1}$$

Its gradient $\beta_1$ and intercept $\beta_0$ can be estimated by means of linear regression (cf. solid red line in Figure 2a), yielding $\beta_1 = -0.394$ and $\beta_0 = -4.142$ in dB.

In order to verify that the standard deviation $s$ of $\sigma^0$ is constant and independent of $\theta$, we calculated the standard deviation per $\theta_w$-bin, and—as can be seen in Figure 2b—these standard deviations are very similar across the whole incidence angle range, with a very small gradient of $-0.008$ and a small (meta) standard deviation of 0.33 dB.

This allows us to fit the linear model for the backscatter as function of the incidence angle $\theta$, with the corresponding standard deviation computed by taking the square root of the sum of squared errors $(SSE(\sigma_w^0))$ divided by the number of data points $(n)$, adjusted for the degrees of freedom of the model (see [48], chapter 3). Putting everything together, the globally applicable flood-backscatter PDF $p(\sigma^0|F,\theta)$ is defined for a given incidence angle $\theta$ (in °) by the following equation:

$$p(\sigma^0|F,\theta) \quad = \quad n(\sigma^0|\,\mu_w(\theta), s_w^2) \tag{2}$$

$$\text{with}$$

$$\mu_w(\theta) \quad = \quad -0.394\,\theta - 4.142\,[\text{dB}] \tag{3}$$

$$s_w \quad = \quad \sqrt{\frac{SSE(\sigma_w^0)}{n-2}} = 2.75\,[\text{dB}]. \tag{4}$$

### 2.3.2. Non-Flood Backscatter Probability Distributions

Our method's purpose is the detection of water surfaces over normally "dry" land surfaces. Hence, a robust a priori knowledge of the local backscatter response under normal, non-flooded conditions is essential. The recorded radar signal over natural land surfaces consists of temporally constant, e.g., soil and bedrock composition, and sensor-related parameters, and variable factors, such as soil moisture and vegetation conditions. Based on the surface characteristics and the climatic condition, the local backscatter time series $\sigma^0(t)$ usually shows a specific periodic, or harmonic, behaviour called *seasonality*.

For the description of the backscatter's seasonality, we use a harmonic model (Equation (5)), following the approach of [21]. $t_{doy}$ is the day-of-year derived from the actual acquisition time $t$ of the radar measurement by applying the day-of-year conversion $t_{doy} = doy(t)$. $C_i$ and $S_i$ represent the harmonic coefficients/parameters, and $\bar{\sigma}^0(t_{doy})$ is the *expected radar backscatter* at $t_{doy}$. The first cosine coefficient $C_0$ equals $\bar{\sigma}^0$, which is the average radar backscatter, and the first sine coefficient $S_0$ reduces per definition to zero:

$$\bar{\sigma}^0(t_{doy}) = \sum_{i=0}^{k}\{C_i\cos(i\,\nu) + S_i\sin(i\,\nu)\} =$$

$$= \sum_{i=1}^{k}\{C_i\cos(i\,\nu) + S_i\sin(i\,\nu)\} + \bar{\sigma}^0 \tag{5}$$

$$\text{with} \quad \nu = \frac{2\pi}{365}\,t_{doy}$$

Based on each pixel's backscatter time series, the $2k+1$ harmonic parameters are computed via a linear least-squares estimation:

$$||A\,x - l||^2 \rightarrow \min \tag{6}$$

where $A$ is the Jacobi matrix, $l$ the observation matrix (i.e., the pixel's time series $\sigma^0(t)$), and $x$ the matrix containing the unknowns, defined as:

$$
A = \begin{bmatrix} 1 & \sin(\nu_1) & \cos(\nu_1) & \dots & \sin(k\,\nu_1) & \cos(k\,\nu_1) \\ \vdots & & & \ddots & & \\ 1 & \sin(\nu_n) & \cos(\nu_n) & & \sin(k\,\nu_n) & \cos(k\,\nu_n) \end{bmatrix},
$$

$$
l^T = \sigma^0(t) = \begin{bmatrix} \sigma^0_{t_1} & \dots & \sigma^0_{t_n} \end{bmatrix},
$$

$$
x^T = \begin{bmatrix} S_0 & C_0 & S_1 & C_1 & \dots & S_k & C_k \end{bmatrix}
$$

(7)

$A$ has the shape $(n, 2k+1)$, where $n$ is the number of measurements in the time series $\sigma^0(t)$ and $k$ is the chosen order of the harmonic model (and $S_0 = 0$, $C_0 = \overline{\sigma^0}$). As suggested by [21], we set $k$ to 3, which is enough to reproduce oscillations caused by seasonal processes at time scales of $\sim$4 months. Higher $k$-values would imply modelling processes occurring at shorter time scales, and hence, incorporate effects from outliers and short-term events, including floods.

Similar to water bodies, radar backscatter from land surfaces depend on the incidence angle $\theta$, though with a generally lower impact. Nevertheless, the backscatter modeling over land is more demanding, as the impact's strength varies strongly with the particular land cover type. We argued that water can be modelled globally without localised parameters, since we can presume that they have a globally uniform behaviour in the C-band SAR perspective. This is not valid for land pixels, which require localised parameters on the backscatter behaviour, owing to the various land and vegetation surface characteristics.

To avoid that variations caused by vegetation- or soil-induced seasonality are confused with incidence or azimuthal angle effects, the systematic impact of the observation geometry has to be eliminated. A normalization step prior to the estimation of the harmonic parameters as done for ASAR by [25] is not applicable for Sentinel-1. Fortunately, we can exploit the self-repeating orbit geometries and the constant incidence angle values $\theta_\rho$, and we estimate the desired harmonic parameters separately for each local relative orbit $\rho$. Ultimately, we can forward the image and parameters as non-normalised backscatter values to the flood model.

Analogous to above, we model the local pixel's likelihood for non-flooded conditions—with a generic PDF $p(\sigma^0|NF, t, \rho)$ for a given point in time $t$ and relative orbit $\rho$—as a Gaussian PDF $n\left(\sigma^0|\,\mu_l(t, \rho), s_l(\rho)\right)$. In particular, the expected backscatter value from the harmonic model $\overline{\sigma}^0_\rho(t_{doy})$ acts as mean parameter of the local distribution $\sigma^0_\rho(t)$, given $t$ and $\rho$:

$$
\mu_l(t, \rho) = \overline{\sigma}^0_\rho(t_{doy}) \quad [\text{dB}] \tag{8}
$$

To illustrate Equation (8), Figure 4e in Section 3 plots an example backscatter time series $\sigma^0_\rho(t)$ from descending orbit $\rho = \text{D080}$ over our Greek study site, together with the estimates for $\overline{\sigma}^0_\rho(t_{doy})$ and the residuals between them.

Furthermore, here, the standard deviation is inferred from the time-independent $SSE(\sigma^0_\rho)$ of the residuals between the pixel's actual time series $\sigma^0_\rho(t)$ (from the datacube) and the expected values $\overline{\sigma}^0_\rho(t_{doy})$ (from the harmonic model), divided by the model's degrees of freedom:

$$
p(\sigma^0|NF, t, \rho) \quad = \quad n\left(\sigma^0|\,\mu_l(t, \rho), s_l^2(\rho)\right) \tag{9}
$$

with

$$
\mu_l(t, \rho) \quad = \quad \overline{\sigma}^0_\rho(t_{doy}) \quad [\text{dB}] \tag{10}
$$

$$
s_l(\rho) \quad = \quad \sqrt{\frac{SSE(\sigma^0_\rho)}{n - (2k+1)}} \quad [\text{dB}] \tag{11}
$$

### 2.4. Bayesian Flood Mapping

Reference [25] computed a water probability for a backscatter measurement from residuals against the mean value of the given PDFs. We expand this approach and compute such a probability based on the above backscatter distributions for flood *and* non-flood.

An incoming Sentinel-1 image is forwarded as a pixel array to the flood mapping algorithm and defines the day of the acquisition $t_{doy}$ and the relative orbit $\rho$. With incidence angle values $\theta_\rho$ from the 2020 mean $\theta$ data, we are able to construct the flood PDF $p(\sigma^0|F)$, and with the harmonic local parameters of orbit $\rho$ and the date $t_{doy}$, we are able to construct the non-flood PDF $p(\sigma^0|NF)$ (for the sake of brevity, from here on, we will omit the conditioning variables, except the class labels $N$ and $NF$). These distributions can be set into relation with new backscatter measurements $\sigma^0$ to assign them to either of the classes flood ($F$) or non-flood ($NF$). Given the class-specific PDFs (or likelihoods), the two *posterior probabilities* $P(F|\sigma^0)$ and $P(NF|\sigma^0)$ of class membership can be inferred using *Bayes' theorem*:

$$P(F|\sigma^0) = \frac{p(\sigma^0|F)\,P(F)}{p(\sigma^0)} \tag{12}$$

$$P(NF|\sigma^0) = \frac{p(\sigma^0|NF)\,P(NF)}{p(\sigma^0)} \tag{13}$$

The denominator $p(\sigma^0)$ is referred to as the *evidence* and serves as a normalization factor to scale the posterior probabilities between 0 and 1 for each sample $\sigma^0$:

$$p(\sigma^0) = p(\sigma^0|F)\,P(F) + p(\sigma^0|NF)\,P(NF) \tag{14}$$

where $P(F)$ and $P(NF)$ are called *priors* and represent the a priori probability of a pixel belonging to a certain class. In the Bayesian framework, these priors could be used to integrate information available before (i.e., a priori) the actual observation, e.g., from historical flood records or run-off models, into the inference. In general, we have no such information, which is reflected by choosing an uniformed prior distribution, assigning for the priors $P(F) = P(NF) = 0.5$ and achieving an equal weighting.

Inserting the posterior probabilities defined in Equations (12) and (13) into the Bayes' decision rule:

$$c = \arg\max_j P(\omega_j|\sigma^0) \tag{15}$$

This yields the most probable class $c$ from the overall class set $\omega = \{\omega_1, \omega_2\} = \{F, NF\}$. Figure 3 shows a graphical illustration of the Bayes flood mapping procedure for an exemplary backscatter observation with respect to the distributions.

The advantages of this approach are that it not only produces the local optimal threshold for separating both classes, but that it also establishes a measure of uncertainty, described by the so called *conditional error*. For each sample $\sigma^0$, one can define the conditional error $P(error|\sigma^0)$ as follows:

$$P(error|\sigma^0) = \min[P(F|\sigma^0), P(NF|\sigma^0)]. \tag{16}$$

$P(error|\sigma^0)$ is the posterior probability that the specific observation $\sigma^0$ was generated by the class not chosen by the Bayes decision rule, which can range from very certain $(P(error|\sigma^0) = 0)$ to tossing a coin $(P(error|\sigma^0) = 0.5)$. As such, it is an inverse measure for confidence in the classification.

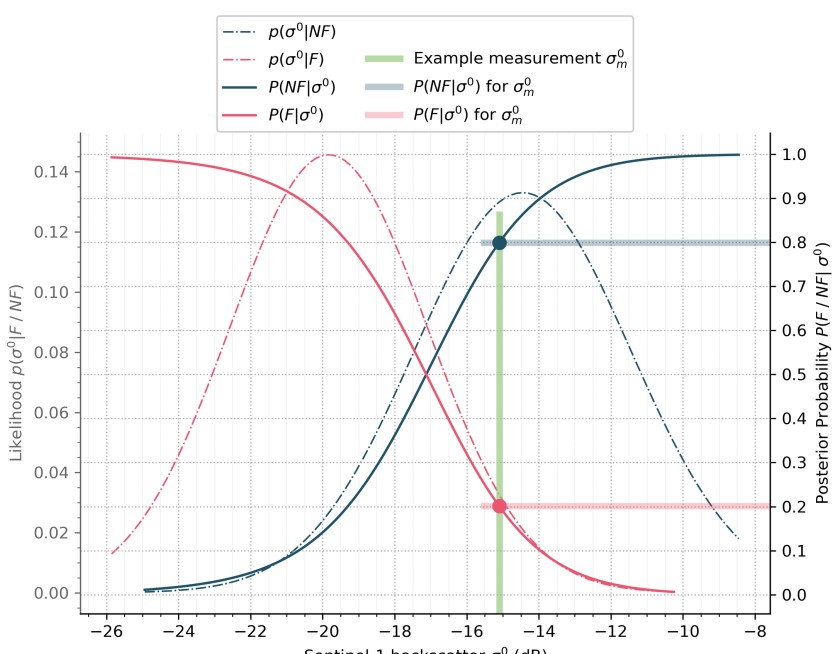

**Figure 3.** Bayesian flood mapping procedure over one pixel for an exemplary backscatter measurement, including the probability density functions (PDF; $p(\sigma^0|F/NF)$ and posterior probabilities $\left(P(F/NF|\sigma^0)\right)$ of the two classes flooded (*F*) and non-flooded (*NF*). This example illustrates a challenging situation with relatively close local parameters $\left(p(\sigma^0|F): \mu_w = -19.83, s_w = 2.73 \mid p(\sigma^0|NF): \mu_l = -14.43, s_l = 2.99\right)$. The marked backscatter observation at -15.1 dB has a probability of 0.8 to belong to the class non-flood (*NF*), with an uncertainty of 0.2.

### 2.5. Detection of No-Sensitivity

The interaction of C-band microwaves with the land surface is in general complex and there a several situations where Sentinel-1 CSAR observations are insensitive to flood conditions for physical, geometric, or sensor-side reasons. With our statistical model parameters built from the multiyear Sentinel-1 datacube, we have a powerful tool at hand to identify such adverse conditions. In the following, we outline our methods to identify locations and observations for which the Bayes model does not allow for a robust decision between flood and non-flood, and thus, our algorithm is ill-posed. The implemented set of masks are widely overlapping, but each one addresses particular aspects of non-sensitivity, and thus, increase the algorithm's robustness, ultimately also aiming at global application.

#### 2.5.1. Masking of Exceeding Incidence Angles

The Sentinel-1 IW mode scans Earth with side-looking viewing angles between 29°–46°. Consequently, flat areas feature incidence angles (IA) only from within this range, whereas IA exceeding it stem from sloped surface, and hence, are only found in rugged terrain. Naturally, water surfaces are observed at all times under IA from this flat range and our water distributions—leading to the regression parameters in Equation (3) and Equation (4)—could only be defined for this limited $\theta$ domain (cf. with Figure 2).

To preclude any flood decision over areas with exceeding IA, we set hard $\theta_\rho$ thresholds before we apply the Bayes model to an incoming Sentinel-1 scene. To allow a decision on flooded conditions in areas on the onset of hills (while keeping the extrapolation moderate), we extend the acceptable $\theta_\rho$ range by a ~10% buffer and relax the initial range to 27°–48°, and obtain the following for the *incidence angle mask*:

$$m_{ia}(\theta_\rho) : \theta_\rho < 27° \ \& \ \theta_\rho > 48° \tag{17}$$

With this, we obtain an a priori mask dependent on the relative orbit, and all pixels with $\theta_\rho$ values outside this range are mapped as unclassified.

### 2.5.2. Identification of Conflicting Distributions

The key indicator driving our algorithm to decide weather a Sentinel-1 measurements stems from a flooded or non-flooded surface is the sharp decrease in backscatter when a normally dry pixel is water-covered. This implies that the pixel has, during normal conditions, higher backscatter values than a respective water surface, equivalent to when the local distribution is overall higher than the respective water distribution. For situations where this is not true, our Bayes decision model is insensitive to flood conditions and cannot be used. Typical locations where this can appear are asphalt surfaces along highways or airstrips, salt panes, or arid san- and bedrock areas, which can be summarised in the SAR perspective as water look-alikes.

Fortunately, with our local and water distributions built from Sentinel-1 backscatter samples, we can determine a priori such ambiguous locations, with respect to the relative orbit $\rho$ and the day-of-year $t_{doy}$. Whenever the local distribution is not distinguishable from the the water distribution, we declare the model insensitive. In particular, we rule out every configuration where the mean of the local distribution is lower than the mean plus one half standard deviation of the water distribution:

$$m_{cd}(\rho, t, \theta_\rho) : \mu_l(t, \rho) < \mu_w(\theta_\rho) + \tfrac{1}{2} \cdot s_w \qquad (18)$$

The choice of this (conservative) threshold not only secures that we exclude locations that have on average backscatter values lower than water, but also rules out configurations where the two distributions share a considerable overlap and the Bayes model becomes arbitrary. From Equation (4), the threshold $m_{cd}$ is always $\mu_w(\theta_\rho) + 1.38\,dB$ and is illustrated as a parallel dotted line to the water's backscatter regression line in Figure 2. With this, we obtain an a priori mask dependent on the relative orbit and the day-of-year, and all pixels with non-separable distributions are mapped as unclassified.

Although many critical cases will be caught by the Bayes uncertainty mask (see Section 2.5.4), this *conflicting distribution mask* $m_{cd}$ profits from the high spatial quality of the multiyear parameters that have a much reduced noise level compared to an individual Sentinel-1 IW scene, and thus, suppresses speckled classifications in noisy SAR image sections when the Bayes decisions are tight.

### 2.5.3. Removal of Measurement Outliers

Our flood algorithm decides between normal and flood conditions on the basis of distributions sampled in multiyear time series. When an incoming Sentinel-1 IW image contains extreme values, i.e., statistical outliers, those measurements are not properly represented by our model's probabilities and a Bayes decision is not meaningful. Independent from the reason (either physical features on the ground, sensor-side image artifacts and energy overflows, or noise and speckle), we exclude such extreme image pixels and mask all values outside three standard deviations of the local distributions:

$$\begin{aligned} m_{out}(\sigma^0, \rho, t) : \sigma^0 &< \mu_l(t, \rho) - 3\,s_l(\rho) \ \& \\ \sigma^0 &> \mu_l(t, \rho) + 3\,s_l(\rho) \end{aligned} \qquad (19)$$

With this, we obtain an *outlier mask* dependent on incoming image values with respect to the local distribution, and all outlier-pixels are mapped as unclassified.

### 2.5.4. Denial of High Uncertainty on Decision

The Bayes approach yields in addition to the classification flood/non-flood the conditional error as measure for its uncertainty. For certain situations—with backscatter values belonging in all likelihood either to the water or the local distribution—this uncertainty measure is close to zero and we can accept the classification with high confidence. In

contrast, when backscatter values of the incoming image are somewhat between the two distributions—falling into their overlap and no class is much more probable than the other one—the Bayes decision is very uncertain and the classification is not meaningful. The maximum value for the conditional error is per definition 0.5, and we define a threshold of 0.2 for an acceptable and meaningful decision, reflecting a 4:1 probability that the assigned class is correct:

$$m_{cert}(\sigma^0) : 0.2 < P(error | \sigma^0) \tag{20}$$

With this, we obtain an *(un-)certainty mask* dependent on the Bayes decision, based on information from the parameters as well as the actual measurement. All pixels where the acceptable certainty is not reached are mapped as unclassified.

### 2.6. Postprocessing

SAR offers an advantageous observation principle when it comes to flood mapping (high sensitivity to water, clear view through clouds, independence from daylight). This comes with some inherent disadvantages, which might impair the correct flood identification, and necessitate a (mild) postprocessing of our classification results.

### 2.6.1. Morphological Operator

The backscatter measurement of a single Sentinel-1 pixel is composed of a superposition of signals from various different scatterers at a sub-pixel scale. This so-called *speckle*, although physical, appears in SAR images as noise and varies the backscatter over homogeneous targets. As a consequence, single pixels may show a lower backscatter and could be confused with inundation. In order to reduce such false-positive detections, a small *spatial majority filter* is applied to the flood map from the pixel-based Bayes decision. A too large kernel size would increase the risk of underdetection, and based on visual impression, we set the filter's kernel size to 3 pixels.

### 2.6.2. Topography

Another quality-degrading effect comes from the side-looking geometry of SAR observations in form of potential signal distortions in areas of strong topography. Range-Doppler terrain correction may not model sufficiently accurate the illuminated area and this could result in very low and high backscatter along hills and mountain ridges.

To avoid strong over- or underdetection, here, we use the Height Above Nearest Drainage (HAND) index data based on the Shuttle Radar Topographic Mission (SRTM) DEM to mask out areas with strong topography that are distant to water bodies. The HAND index value [49] represents the vertical distance between a DEM cell and the nearest cell of the drainage network. By excluding all pixels featuring a HAND index of 20 m or more, the impact of topography related misclassification can be reduced significantly. Since floods appear predominately in vicinity to local aquifers and take effect only on flat terrain, the risk of missed classification is minimal through this exclusion, as (relatively) low-lying areas remain unmasked.

This HAND mask largely overlaps with the incidence angle mask from Section 2.5.1, as in practice, both are a function of topography. However, while the $m_{ia}$ mask covers areas where our approach is unfit for flood mapping because of the Sentinel-1 SAR geometry, the HAND mask excludes areas where one cannot expect floods in general, from a hydrologic perspective. Effectively, the HAND mask removes artefacts on flattening hilltops, and thus, improves the algorithm's robustness.

### 2.7. Study Flood Event and Reference Data

To test our method, we examine a major flood disaster that struck the Greek mainland in February 2018. This event was already subject to, e.g., the study of [26], and serves our experiment well, as it was well-captured not only by Sentinel-1, but more importantly, also by the (usually rare) reference satellite products with a similarly good temporal overlap.

In 2018, the region of Thessaly (overview map in Figure 4 in Section 3) was subject to ongoing and unprecedentedly intense rainfall, causing rivers to overflow and inundating farmland and settlements. A weather station near the village of Zagora recorded 676 mm of rain between 21 and 26 February, including 209 mm in only 24 h on the 26th [50]. The most affected area spans about 50 km in the north-west of the Thessalian plain between the cities of Trikala and Larissa, accommodating dense agriculture with non-irrigated and permanently irrigated farmland on fertile soils. The plain is widely surrounded by mountain ranges and is discharging into the Aegean Sea via the Pineios River and its tributaries. This drainage basin had been frequently flooded in history [51] and is still flood-prone nowadays, hit again by a severe flood of the Pineios river in February 2018.

After its peak around 26 February 2018, two Sentinel-1 images recorded the situation on 28 February 2018 at 04:39 and 16:31 from two overpasses in the orbit tracks with the relative orbit number D080 (descending direction) and A175 (ascending). As validation references from the same date, we could collect one flood delineation map from Copernicus Emergency Management Service (CEMS) Rapid Mapping (https://emergency.copernicus.eu/mapping/, accessed on 13 January 2022), and one (almost cloud-free) Sentinel-2 multispectral-optical acquisition. The latter was converted to a flood map by calculating the Normalised Difference Water Index (NDWI) and setting the threshold of 0.6 (and by neglecting the very small permanent waters), following [52]. The reference data for the evaluation in Section 3 are detailed in Table 1.

**Table 1.** Reference datasets over 2018's flood event in Thessaly.

| |
|---|
| **Copernicus EMS** (COSMO-SkyMed) |
| *28 February 2018 at 04:20* |
| *source file:* EMSR271_02FARKADONA_01DELINEATION_MAP_v1_vector.zip |
| **Sentinel-2** (flood map through NDWI > 0.6) |
| *28 February 2018 at 09:20* |
| *source file:* S2B_MSIL1C_20180228T092019_N0206_R093_T34SEJ_20180228T144732.SAFE |

## 3. Results and Discussion

This section comprises in-depth analyses of the parameter sets input to our Sentinel-1 Bayes flood classification algorithm, demonstrated for the Thessaly study site. We first examine the local distribution's expected mean values from the harmonic model, and how they relate to local SAR seasonality and to the global water reference. Then, we analyse the Bayes flood model's a priori sensitivity for the particular flood event on 28 February 2018. The third part then assesses the actual flood map results against the two external reference maps, and discusses the performance in relation to land cover and applied masks.

### 3.1. Non-Flood Parameters from Harmonic Analysis

When preparing the inputs to our Bayes flood model, the reference water distribution is established just through the globally applicable linear relationship from Equation (1). In contrast, the (non-flood) local distribution is defined for each pixel and for each relative orbit covering the monitored area. Figure 4 maps over our study site in Thessaly the mean parameter of the local distribution of the descending orbit $\rho = D080$ for four selected days of the year (*doy*). These means are the expected values $\overline{\sigma}_\rho^0(t_{doy})$ obtained from the harmonic model in Equation (5) and form a synthetic backscatter image describing the average C-band SAR signature for the specific *doy*. In our set of example images, one can clearly see the seasonal development of the backscatter, with distinct patterns particularly over the agriculture of the Thessalian plane.

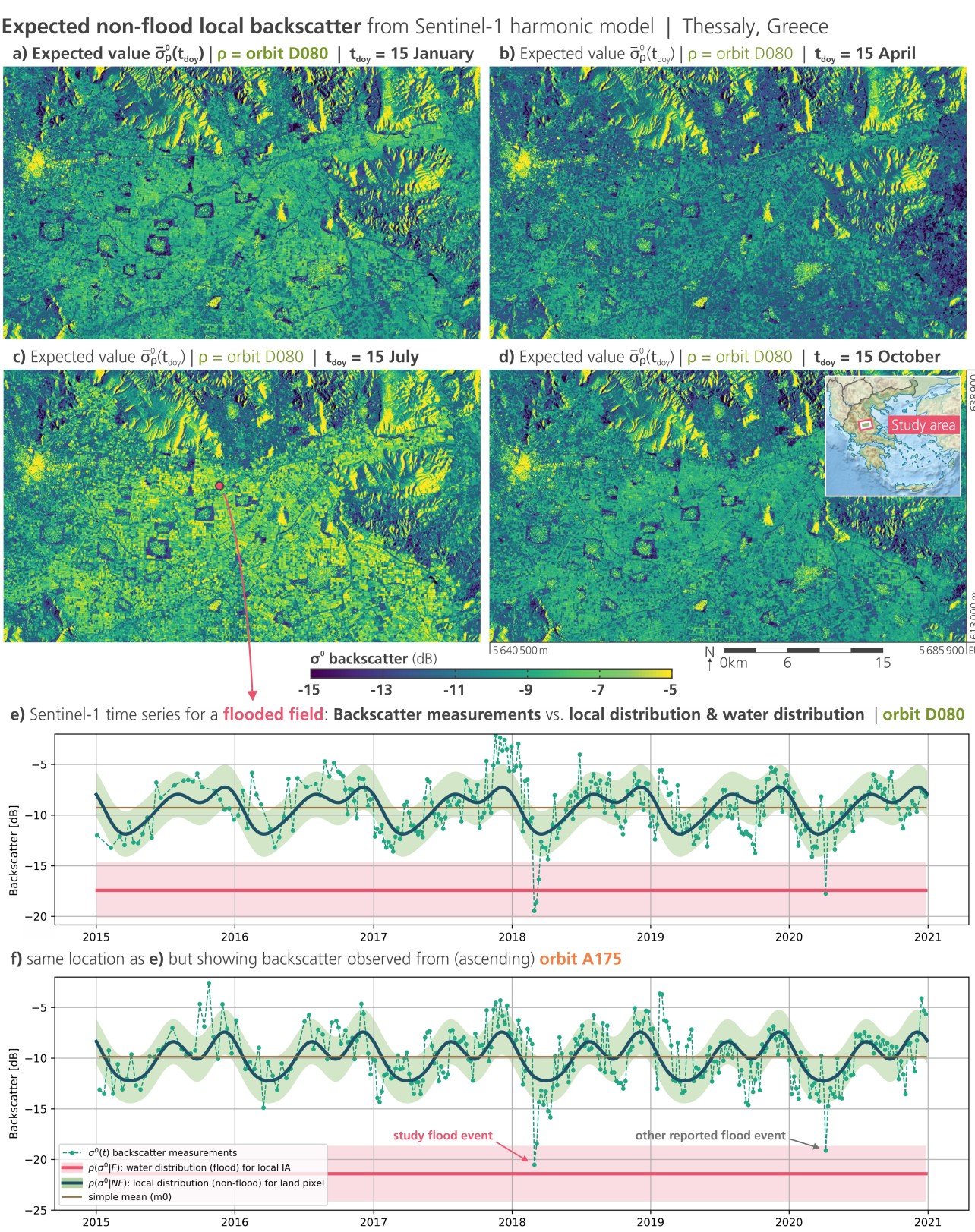

**Figure 4.** Over Thessaly, Greece, (**a**–**d**) examples from the local distribution dataset for four selected days of the year (*doy*), i.e., the expected backscatter $\overline{\sigma}^0_\rho(t_{doy})$ for relative orbit $\rho = D080$ resulting from the harmonic analysis defined by Equation (5). Additionally, (**d**) contains the Equi7Grid-coordinates, and the location of the study site as a minimap. (**e**,**f**) For two relative orbits, example temporal plots showing (expected) local distribution over observed backscatter and the (locally static) water distribution.

Figure 4a shows for 15 January in the west the city of Trikala with highest backscatter ($>-6\,\mathrm{dB}$), and lowest backscatter ($<-13\,\mathrm{dB}$) over roads, rivers, hill slopes, and peculiar rectangle-shaped grassland-formations around the smaller settlements. In between, covering the largest area, the different farms/fields show a typically contrasted signature in this SAR imagery. Furthermore, very prominent are the backscatter signatures following the topography, with their distinct alteration between slopes facing towards and against the sensor viewing direction, which is for this (descending) orbit from east to west. Figure 4b shows for 15 April a general much lower backscatter level, also with reduced contrast during this early stage of the vegetation cycle. Completely in opposite to this, Figure 4c on 15 July shows much higher $\overline{\sigma}^0_\rho(t_{doy})$ values for the majority of the farm plots, while many other features from the January image are recognized. The overall backscatter level then again decreases until 15 October, shown in Figure 4d. This seasonal behavior is well exemplified by the time series of one selected pixel, which are plotted for the two relevant Sentinel-1 orbits in Figure 4e,f, together with the local distribution's mean (the expected $\overline{\sigma}^0_\rho(t_{doy})$), the local distribution's standard deviation ($s_l(\rho)$), and the water distribution ($\mu_w(\theta)$ and $s_w$) for the incidence angle ($\theta$) for this pixel and respective orbit. While the water distributions are constant over time per definition, the local distributions follow an annual cycle, most of the time approximating well the actual time series. The two plots of the expected backscatter from opposed orbit directions describe both the local strong bimodal seasonality that distinguishes well the constant simple temporal mean, but with some narrow differences between the orbits related to azimuthal SAR effects and overpass timing (morning vs. evening). Furthermore, they show the different distances between local and water distributions, directly affecting our method's sensitivity (cf. Figure 5e,f)).

A couple of times, the example's residuals between actual and expected values are larger and "leave" the standard deviation. Some high peaks may be attributed to crop rotation and relatively wet soil moisture conditions, but the largest residuals are negative, and are of particular interest to our study: in fact, the large drop much below the reference water distribution in early 2018 corresponds to the reported 28 February 2018 flood event. Accordingly, this pixel will be classified as flooded through the Bayes inference, as discussed later in Section 3.3. We note that such negative magnitudes were also experienced in 2020, when floods were reported in this area, but which are not discussed in this study due to a lack of comparable reference data, and for the sake of brevity.

### 3.2. Bayes Model: A Priori Sensitivity Analysis

In this section, we examine the model's parameters and its a priori sensitivity towards flood mapping for the particular case of the Thessalian flood event. Figure 5 shows for the two Sentinel-1 orbit-overpasses available on 28 February 2018 the parametric baseline, with the IA-dependent mean of water distribution (Figure 5a,b), the time-dependent mean of the local distribution (Figure 5c,d), and the $\sigma^0$ difference between them (Figure 5e,f). Finally, Figure 5g,h maps the 6-year standard deviation of the local distribution, which is provided to the model as a locally constant parameter, and is in general a function of the land cover, with low backscatter variation over cities, roads, and the vegetated Pineios river banks, and with a diverse pattern following the agricultural plots.

The difference maps in Figure 5e,f are the direct result of the four respective maps above and are of special interest, as they reflect to a great part the model's sensitivity to floods. A large difference between the distribution is a favourable condition for well-secured Bayes decisions between flood and non-flood (cf. Equation (15) and Figure 3). One can see that the configuration in the afternoon at 16:31 for the orbit $\rho = A175$ is more favourable, whereas the morning overpass of orbit $\rho = D080$ features overall smaller differences and significant more areas that fall below the 1.38 dB-limit (marked in blue, cf. Figure 2). These areas will be masked during the flood mapping as pixels with conflicting distributions, through application of Equation (18).

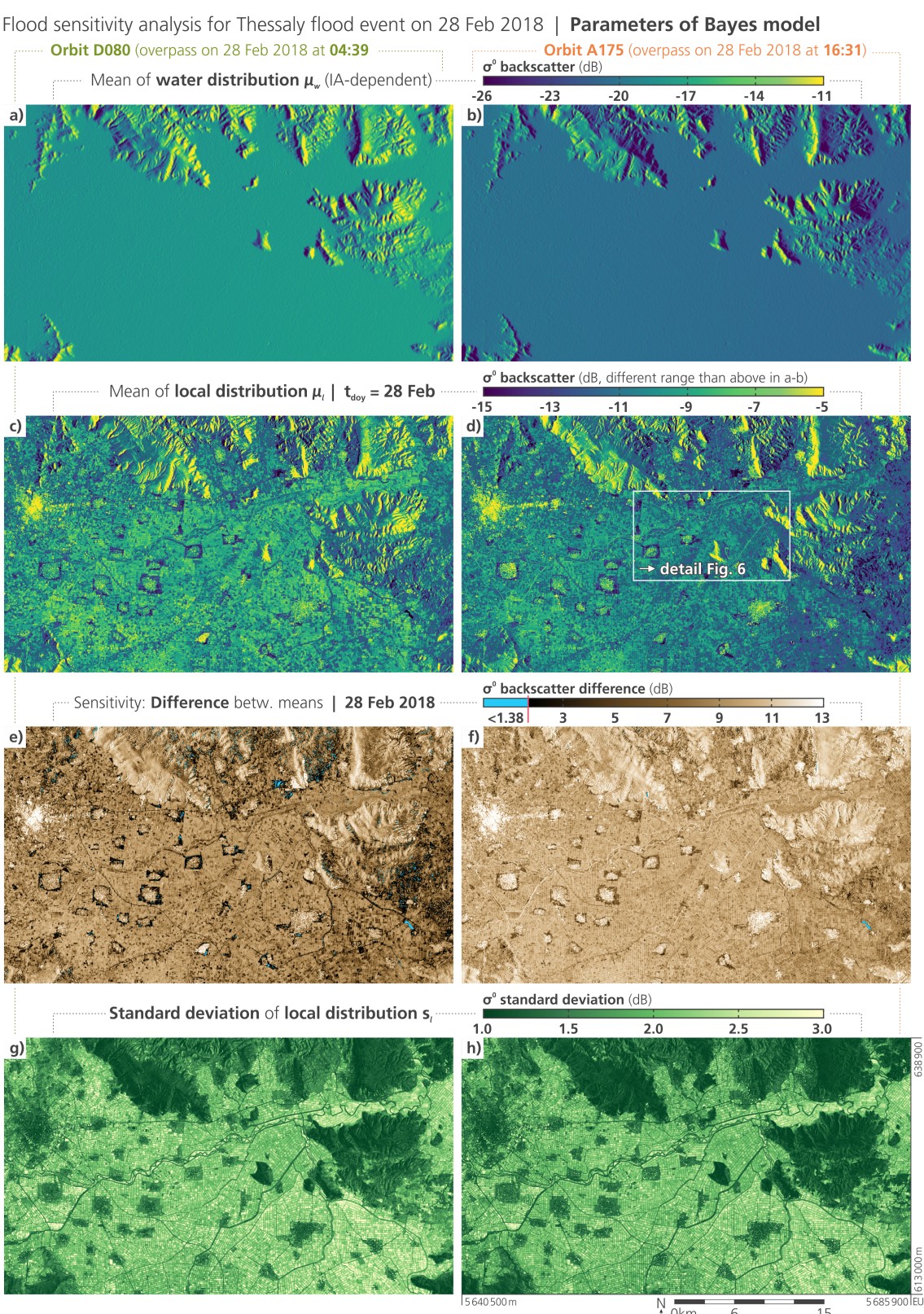

**Figure 5.** Over the complete study site in Thessaly, the Bayes flood algorithm spatial parameters for 28th February and Sentinel-1's two overpasses in 2018. (**a**,**b**) The mean of water distribution $\mu_l(\theta)$ as a function of incidence angle $\theta$. (**c**,**d**) The mean of local distribution $\mu_l(\rho, t)$ from the harmonic model. (**e**,**f**) The differences between them indicating the flood model's sensitivity for this configuration, and highlighted in blue the minimum difference evoking masking (cf. Section 2.5.2). (**g**,**h**) The standard deviation of the local distribution from Equation (11).

On these grounds, Figure 6 arranges a zoomed-in analysis of the flood situation at 16:31, and how the Bayes flood posterior probability $P(F|\sigma^0)$ identifies inundations. Figure 6a displays the actually observed Sentinel-1 scene from this overpass from ascending orbit 175, with—most strikingly—a strong and clear impression of water surfaces dominating this sector. When comparing this image with the expected backscatter for this day-of-year from the harmonic model in Figure 6b, our flood algorithm's principle as a means of change detection becomes obvious. While the presumably non-flooded areas show in both datasets similar patterns, the surroundings of the Pineios river (in the east–west direction) and its smaller tributaries and canals show large and distinct zones of very low backscatter, clearly marking flood bodies.

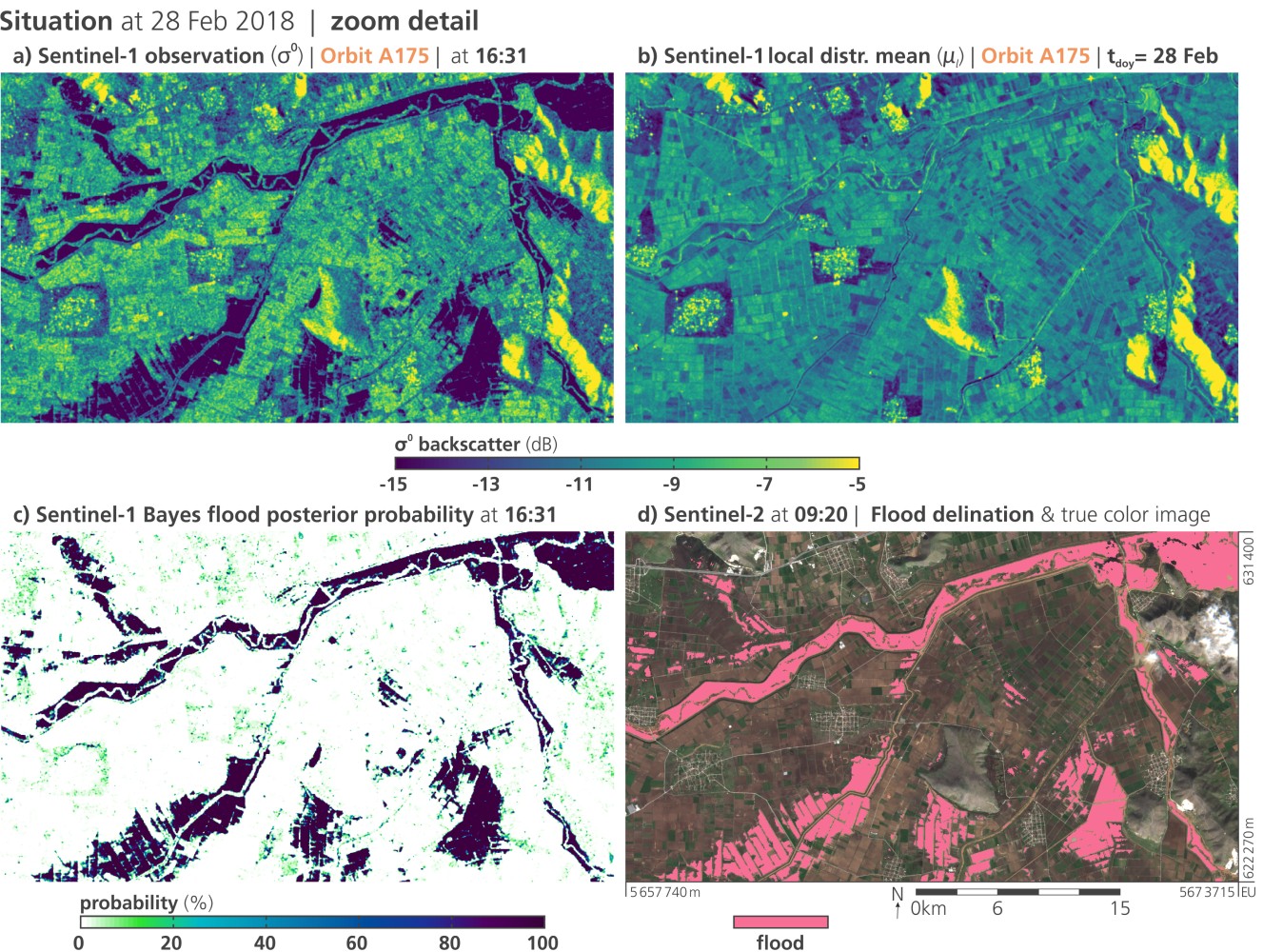

**Figure 6.** Detailed analysis of Thessaly flood situation on 28 February 2018, comparing (**a**) the backscatter image observed by Sentinel-1 from ascending orbit 175, (**b**) the expected backscatter $\overline{\sigma}_\rho^0(t_{doy})$ for 28 February, (**c**) the resulting Bayes flood posterior probability $P(F|\sigma^0)$, and (**d**) the flood map from Sentinel-2 NDWI on the preceding morning with the corresponding RGB image in the background.

Integrating these data with the respective water distribution, and applying the equations from Section 2 (cf. Equation (12)), we obtain i.a. the Bayes posterior probability for flood that is displayed in Figure 6c. This flood probability map already bears a strong resemblance with the reference flood map built from the Sentinel-2-based NDWI image shown in Figure 6d that captured the flood situation on that day at 09:20. The comparison of our SAR data with this optical image also aids the interpretation of the local land cover with its farming plots, the forested hills, and settlements.

The (dark) box-shaped zones of bare soils and sparse vegetation in the vicinity of most villages pose a typical challenge to SAR-based flood mapping, as they typically generate low backscatter, and thus, may have a higher chance to be mislabelled as water bodies. Here, our parameter set shows its strength, as the regular seasonal dry conditions—that exhibit lower backscatter than other times of the year—are modelled by the harmonic analysis and will be interpreted appropriately by means of the Bayes decision. Consequently, this year's slight increase in the observed backscatter—versus the (expected) harmonic mean—leads to a small flood probability of ∼20%. Although not close to zero, it is still rather low and will not be labelled as flood.

Another observation in Figure 6a is that the meandering Pineios river—which can be recognised in both the observed and the expected SAR images—exhibits higher backscatter than the standing water. The river has in this section a varying width of 15 m to 25 m, and at the 20 m resolution, the dense vegetation lining the river edges are not resolvable and lead to a mixed SAR signal. Eventually, this renders those river pixels insensitive to our algorithm (while technically, they should not be classified as flood in the first place). As one can see, the Sentinel-1 NDWI-flood map offers more precision in this respect, with flood bodies fitting closely to the river line.

### 3.3. Flood Map Results and Evaluation

In the following, we display and discuss the flood mapping outputs resulting from our Bayes method, first with the scope of the entire site in Figure 7, and then with zoomed-in focus with respect to two reference datasets in Figure 8, each for both Sentinel-1 overpasses on 28 February 2018.

Figure 7a,b shows the observed Sentinel-1 images overlaid by the outline of our obtained flood classification. Overall, the huge extent of the flood crisis becomes evident, with about 5% of this ∼1200 km$^2$-large map under water. The observations were captured after the intense rainfalls and the flood's local peak, and the Sentinel-1 data confirm this by mapping a regress in total flood extent from 67 to 55 km$^2$ between the morning and afternoon overpass, spanning a ∼12 h window. This progression is supported by the two 5 h–apart reference datasets from CEMS and Sentinel-2 in Figure 7g,h, as the later map indicates that the floods slightly withdraw from the higher elevated areas far from the main river.

Figure 7c,d maps the classification uncertainty by the Bayes decision with respect to conditional error $P(error|\sigma^0)$ defined by Equation (16). The higher uncertainty for the data configuration of orbit D080 in the morning is apparent, owing mostly to the larger overlap of the local and water distributions, where the latter features higher/closer backscatter due to the orbit's smaller incidence angles. The uncertainty patterns, which follow chiefly challenging land cover and lines of mixed signals (e.g., along the rivers), can be directly related to the lower sensitivity shown in Figure 5e,f. Consequently, and also reassuringly, the generated masks defined in Section 2.5 and shown in Figure 7e,f are broader for orbit D080. The three non-topographic masks (which deal with the Bayes configuration; cf. Sections 2.5.2–2.5.4) label for D080 4.9% of the total area, but only 1.8% for A175. Interestingly, it appears that the stronger masking (which results from the orbit's lower sensitivity) mainly impact the non-flooded class, as the masks reduce the flood class extent by 2.8% for both orbits alike.

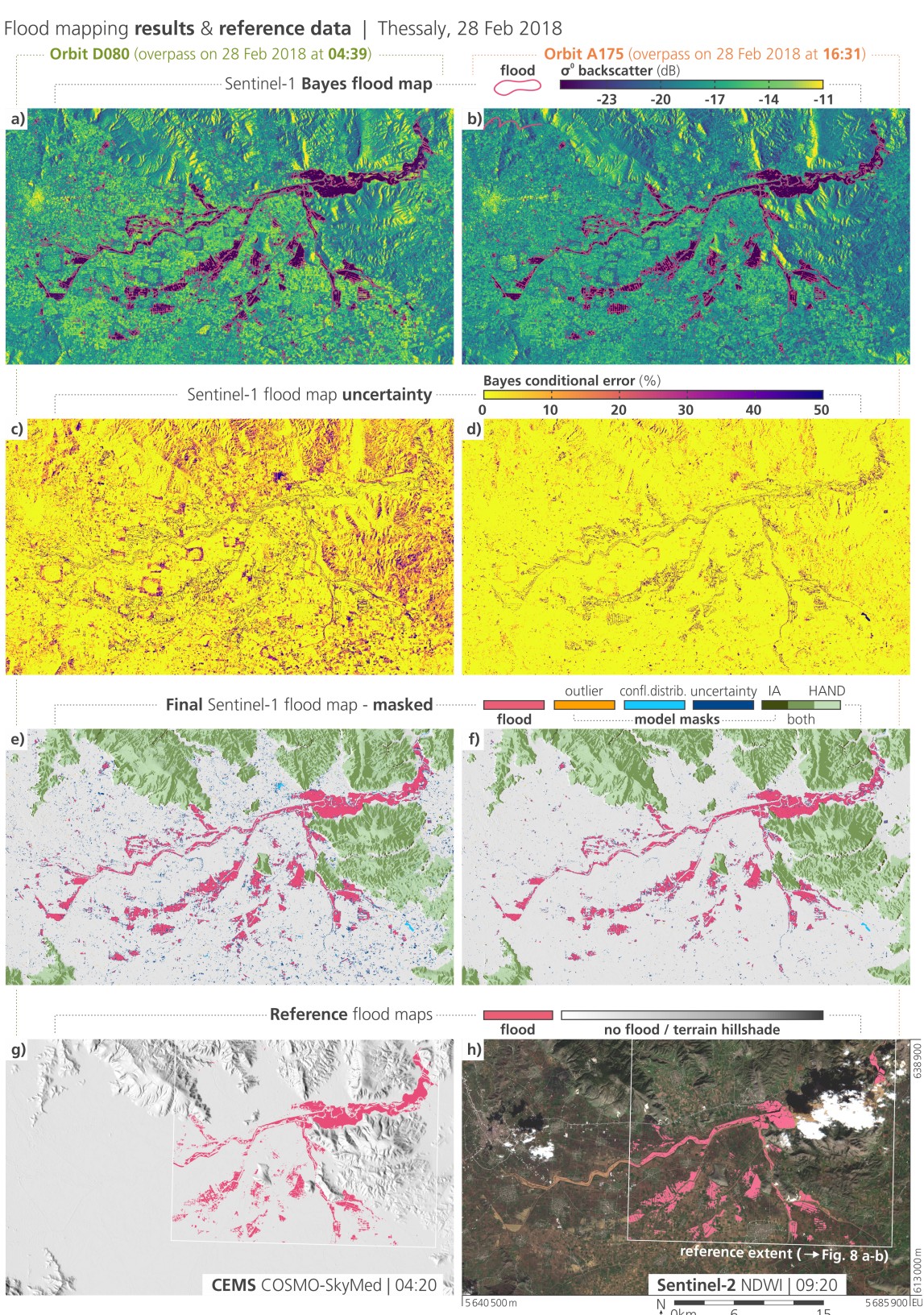

**Figure 7.** Array of Bayes flood mapping results versus reference flood maps from (**g**) CEMS and (**h**) Sentinel-2. (**a**,**b**) Sentinel-1 backscatter images from two overpasses on 28 February 2018, overlaid with the outline of the Bayes flood classification. (**c**,**d**) Uncertainty values (i.e., conditional errors) of the Bayes decision. (**e**,**f**) Final Bayes classification results after postprocessing and masking (subject to evaluation).

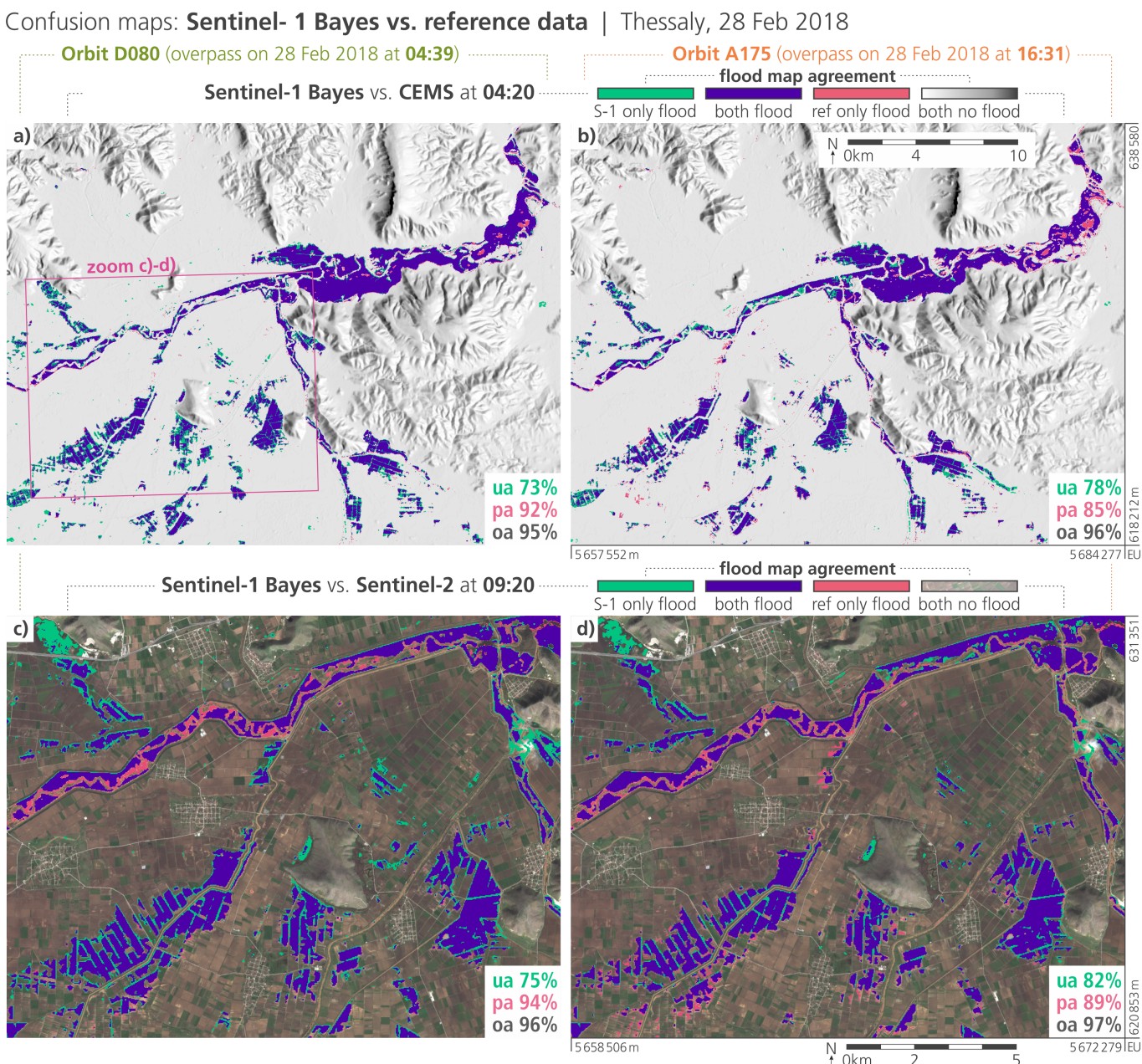

**Figure 8.** Confusion maps between the final Sentinel-1 Bayes flood mapping results and the reference flood maps, from (**a**,**b**) CEMS and (**c**,**d**) Sentinel-2, with respective User Accuracies (UA), Producer Accuracies (PA), and Overall Accuracies (OA). The lower panel shows zoomed-in maps to illustrate spatial detail, but provide accuracy metrics representing the complete area.

The masked and postprocessed flood extents displayed in Figure 7e,f are the ultimate results subjected to our evaluation against the reference datasets. Figure 8 zooms into the domain where the CEMS (Figure 8a,b) and Sentinel-2 (Figure 8c,d) data were available and plots the respective *User Accuracies (UA), Producer Accuracies (PA)*, and *Overall Accuracies (OA)* against the binary reference maps. Please note that the evaluation and statistics were done for the whole reference extent, but Figure 8c,d zooms further into a cloud-free extent of the Sentinel-2 data to illustrate the detail of the classification (and the deviations). Relative to the CEMS reference map, our Sentinel-1 flood maps have a UA of 73% and 75% and a PA of 92% and 85%, respectively, for tracks D080 and A175. Relative to the Sentinel-2 reference map, we obtained higher values of agreement, with a UA of 75% and 82% and a

PA of 94% and 89%, which is an interesting finding, as one could expect rather less than higher agreement with an optically derived flood extent.

Overall, we can assess that our method performs well, since the results agree fully with the reference datasets on the substantial structure and extent of the flood bodies. When looking on Figure 8, major differences with respect to the references appear

1. In the north-east as missed floods (false negatives in red);
2. In the north-west as additional floods (false positives in green);
3. Along the meandering Pineios river as missed floods;
4. Along the outlines of the flood bodies, in particular at the flooded farming plots in the south, as misclassification in both directions.

Cases (1) and (2) are most likely a consequence of the different timings of the satellite observations, as visual inspection of the Sentinel-1 SAR images suggest that these areas are correctly classified by the Bayes method. For example, when comparing Figure 6a with Figure 8d, we can assess confidently that the north-western lobe of the flood there is correct and not a false positive. A similar verdict can be done for the green-labelled flood body in the centre of Figure 8c,d. Consequently, the overall poorer PA for the (later) orbit A175 can be explained by the larger time lag to the reference images. Case (3) is different in this perspective, as already discussed, the resolution power of our 20 m-sampled backscatter data is insufficient here and the finer spatial sampling of the reference maps, i.e., 3 m for CEMS and 10 m for Sentinel-2, are more reliable in land cover transitions zones. The same issue comes into effect for case (4), where the exact outlines of the flood bodies over the agricultural plots are in dispute. However, here, the question of what is the "correct" flood extent is intermingled again with the issue of the exact observation time, as the flat terrain there let the lateral water extent change quickly during flood progression.

Generally, such *fringes of heterogeneous pixels* (well visible also in the uncertainties in Figure 7c,d trouble our SAR data in both the spatial and temporal dimension. While the former issue is shared typically by all SAR algorithms, i.e., when incompletely flooded pixels do not exhibit specular reflection, the latter challenges particularly our algorithm with its dependence on the temporal signal. Since the non-flooded local probability distribution is generated from the observational time series, higher temporal variability leads to wider local distributions, and ultimately, to higher classification uncertainty (recalling that in Bayesian classifiers, smaller class standard deviations produces good separability). Fortunately, the here-used harmonic model as a seasonally fitted reference, compared to just using, e.g., the median or mean, reduces these deviations for areas where seasonality is apparent. On the contrary, when the local variability is not bound to re-occurring seasonal patterns, e.g., in cases of land cover change or disturbed water reflectivity due to wind, the parameters are not well determined. A related concern here are deviations from seasonal patterns coming from *crop rotation* in agricultural areas. While the variance increases only little, the larger concern is the possibility that the expected day-of-year backscatter $\bar{\sigma}^0_\rho(t_{doy})$ is not representative for the actual year (the mean of the local distribution is ill-placed). For instance, Figure 6a,b shows a few fields with deviant expected (mean) backscatter in the north-eastern quarter. Another issue in this respect are *drier-than-normal soils*, which results in a decreased backscatter signal that in extreme cases may be mistaken as floods.

## 4. Conclusions

In this paper, we presented our recent advances in flood mapping with Sentinel-1 SAR data, which produced a novel method that is fit for global and near-real-time monitoring. Operating autonomously from human interaction and reference identification, it yields flood classification and corresponding uncertainty values by distinguishing current SAR imagery from precomputed and localised parameters. The algorithm is centred on an optimised global datacube structure, is parametrised pixel-wise through harmonic time series analysis, and features a priori masking of insensitive areas and observations.

We established our approach on the basis of the monitoring capability of the European Sentinel-1 CSAR mission and its global and long-standing observation scenario (in the light

of the ongoing anomaly of Sentinel-1B, ESA advanced the launch schedule of Sentinel-1C to April 2023). Building upon the stability, frequency, and quality of the provided IWGRDH imagery, we derived seasonality parameters from 2015–2020 time series, and we collected a representative CSAR signature for water bodies. Moreover, we effectively capitalise on the mission's orbit repetition through the per-orbit model parametrisation the usage of static incidence angles, while at the same time minimising the systematic influence of the observation geometry. Data-driven exclusion masks identify situations suffering from unfit parameter configurations, where Sentinel-1 flood mapping is not reliable or even impossible due to physical limitations of the SAR system.

In conjunction with this a priori and pixel-localised flood model calibration, the presented Bayes classification decision engine requires little computational effort, and hence, can be run fast during near-real-time (NRT) flood mapping applications. In fact, the algorithm is already an integral component of the recently launched Global Flood Monitoring (GFM, [53]) component (integrated in the Global Flood Awareness Systems (GloFAS) available at https://www.globalfloods.eu/ (accessed on 25 July 2022)) of the Copernicus Emergency Management Service (CEMS), as one of three independent flood mapping algorithms that are combined within one ensemble decision product. The *GFM ensemble setup* [54] promises robustness and accuracy in global flood monitoring, as the three employed Sentinel-1 algorithms complement each other through entirely different concepts on the the flood decision, and with this, it represents well the current research on automated SAR-based flood detection [55]. The algorithm based on the work of [20] ingests a pair of two recent images and maps changes therein through statistical modelling of backscatter distributions in hierarchical subsets, while the algorithm based on developments by [17] classifies in single images using fuzzy-logic methods and topography-derived indices, with subsequent region growing. In contrast, our here-presented algorithm exploits per pixel the full Sentinel-1 signal history from the datacube, and classifies on the basis of precomputed probabilities for flooded and non-flooded SAR signatures. A big advantage towards NRT-readiness of our approach is that there is no dependency on a recent and congruent precursor satellite image, because the change is detected against a precomputed synthetic image from the harmonic model.

In terms of flood map accuracy, our datacube-based Bayes decision performs reliably, well-aligned with result metrics produced in the literature (e.g., as in the review of [56], or specifically [26]). The obtained flood maps for the Thessaly event in February 2018 are widely congruent with the two available reference maps, with complete agreement on the general flood body structure, and with few notable deviations related to different observation timings and retreating floods. The comparative coarse resolution of our 20 m-sampled datacube may be seen as a shortcoming here, ceding some details along vegetated rivers. However, this mild downsampling, combined with the temporal aggregation by the harmonic synthesis, offers a clear representation of the expected local SAR signature, practically free from noise and speckle. Following this method paper, our group is currently composing a subsequent evaluation study [57]. It examines in depth the flood mapping performance for multiple events on five continents in comparison to maps generated by the CEMS Rapid Mapping activations, with findings that confirm observations made in this study of the 2018 Greece event.

What remains to be tackled by future research is an appropriate handling of **off-seasonality**. This includes effects foremost from crop rotation in agricultures, progressing land cover changes, or extreme soil moisture conditions. In such cases, the harmonic model and the expected backscatter value maybe do not fit the actual non-flood signal, and misinterpretation and false positives can occur. A solution could be adapting the seasonal reference with current observations, e.g., through integrating antecedent Sentinel-1 images and merging them with the local harmonic function through temporal filtering. Furthermore, the impact of the input backscatter time series length in terms of seasonal cycles is to be addressed in the upcoming experiments.

In order to detect **windy conditions** that roughen the flood surfaces, within the GFM project, a first adequate attempt is implemented based on Sentinel-1 data on-hand during runtime. When the current SAR image over regular water bodies is much increased against localised long-term statistics, a wind flag is raised in the surrounding areas. As such, this is independent from auxiliary meteorological data that may be troublesome in global near-real-time operations.

That said, integrating **auxiliary information** is another research direction aiming for further increased accuracy, especially as Bayesian methods are adept in integrating preexisting ancillary information in the labelling process. For floods, data on topography, morphology, or local water body seasonality may be integrated in form of dynamic *prior probabilities*, e.g., in simple Bayesian inference [22], or further integrated in belief networks [24].

Finally, the **masking** of problematic areas opens up a wide field of possibilities to increase robustness, in particular when aiming for automated and global applications. The topical work of [58] focused on the globally applicable generation of a dedicated exclusion mask, where SAR is insensitive to flood/non-flood conditions. In a quite similar approach, they applied time series analyses on the Sentinel-1 datacube to identify problematic land covers and radar geometries (e.g., shadows), and could effectively reduce classification errors.

**Author Contributions:** Conceptualization B.B.-M., V.F. and W.W.; methodology W.W., S.C., F.R., C.N., M.E.T., B.B.-M. and T.M.; software C.N., F.R. and M.E.T.; validation M.E.T. and F.R.; formal analysis M.E.T.; investigation F.R., M.E.T. and B.B.-M.; data curation S.C. and M.E.T.; writing—original draft preparation B.B.-M.; writing—review and editing ALL; visualization B.B.-M. and M.E.T.; supervision B.B.-M. and W.W. All authors have read and agreed to the published version of the manuscript.

**Funding:** This study was funded by TU Wien, with co-funding from the project "Provision of an Automated, Global, Satellite-based Flood Monitoring Product for the Copernicus Emergency Management Service" (GFM), Contract No. 939866-IPR-2020 for the European Commission's Joint Research Centre (EC-JRC), and the project "Flood Event Monitoring and Documentation enabled by the Austrian Sentinel Data Cube" (ACube4Floods), Contract No. 878946 for the Austrian Research Promotion Agency (FFG, ASAP16). The authors acknowledge TU Wien Bibliothek for financial support through its Open Access Funding by TU Wien.

**Data Availability Statement:** The data layers presented in this study are available on request from the corresponding author. Flood data generated by the algorithm is globally available as part of the Global Flood Monitoring (GFM) within the Copernicus Emergency Management Service (CEMS) at https://www.globalfloods.eu/, accessed on 25 July 2022).

**Acknowledgments:** The computational results presented have been achieved using inter alia the Vienna Scientific Cluster (VSC). We would further like to thank our colleagues at TU Wien and EODC for supporting us on technical tasks on maintaining the datacube.

**Conflicts of Interest:** The authors declare no conflict of interest.

## Abbreviations

The following abbreviations are used frequently in this manuscript:

| | |
|---|---|
| ASAR | Advanced SAR onboard Envisat |
| CSAR | C-band SAR onboard Sentinel-1 |
| CEMS | Copernics Emergency Management Service |
| DEM | Digital Elevation Model |
| DOY | Day Of Year |
| EODC | Earth Observation Data Centre for Water Resources Monitoring |
| GRDH | Ground Range Detected High-resolution SAR product |
| GFM | Global Flood Monitoring |
| HAND | Height Above Nearest Drainage |
| IA | Incidence Angle $\theta$ |

| IW | Interferometric Wide Swath mode of Sentinel-1 |
| OA | Overall Accuracy |
| PA | Producer Accuracy |
| UA | User Accuracy |
| PDF | Probability Distribution Function |
| NDWI | Normalized Difference Water Index |
| NRT | Near-Real-Time |
| S-1 | Sentinel-1 |
| SSE | Sum of Squared Errors |
| SIG0 | Sigma Nought backscatter coefficient $\sigma_0$ |
| SAR | Synthetic Aperture Radar |

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
