# Peer review of "Satellite-Based Flood Mapping through Bayesian Inference from a Sentinel-1 SAR Datacube"

_remotesensing, doi:10.3390/rs14153673_

Round 1
Reviewer 1 Report
REVIEW of the manuscript "Satellite-based Flood Mapping through Bayesian Inference from a Sentinel-1 SAR Datacube" by Bernhard Bauer-Marschallinger, Senmao Cao, Mark Edwin Tupas, Florian Roth, Claudio Navacchi, Thomas Melzer, Vahid Freeman, Wolfgang Wagner [Remote Sens. 2022, 1,0. https://doi.org/].
The authors formulate a datacube-based flood mapping algorithm that exploits the Sentinel-1 orbit repetition and a-priori generated probability parameters for flood and non-flood conditions. A globally applicable flood signature is obtained from manually collected wind- and frost-free images. Through harmonic analysis of each pixel’s full time series, a local seasonal non-flood signal comprising the expected backscatter values for each day-of-year is derived. From those pre-defined probability distributions, incoming Sentinel-1 images by simple Bayes inference are classified. The datacube-based masking of no-sensitivity resulting from impeding land cover and ill-posed SAR configuration enhances the classification robustness. The new algorithm was applied to the flood in Thessaly (Greece). The results obtained are better compared to the other algorithms.
The field of work is very current and important for flood information and warning. From the presentation it is clear what is new in the method. However, authors need to answer some more questions before the paper is ready for publication.
Comments
Floods are closely linked to precipitation which is also accurately described by а satellite. The authors should dedicate one paragraph within the Introduction section to the link between precipitation and floods. Can satellite detection of the maximum precipitation also detect the area of the strongest floods?
The Introduction section need to be without subsections. Mostly of subsection 1.2 may be transferred to Method and Results with descriptions of the new algorhitm advantages.
In the version of the manuscript I received, all references in parentheses are blank.
Reviewer 2 Report
No citations are made in the text. The place where they should be made is marked only with "[?]". Also, the list of bibliographic references is completely missing. Please correct this big error.
Subchapter 1.1 seems to be more of a methodology part than an introductory part.
Figures cannot be viewed.
at the introduction it does not present the global and regional context of the works realized on this subject.
Reviewer 3 Report
The figures and the list of references were missing in the file available for the review process.
Both these issues make difficult and incomplete the review of the manuscript.
In addition, some minor points need clarification:
- 1) In the conclusions, it is not clear highlighted and summarized the dataset used to build the proposed approach (in particular, the period of data available for the implementation of the Bayes model)
- 2) It is not clear the quantity of the total area which is excluded by the different masks (at lines 628-631, just three masks are mentioned)
- 3) Line 547: should “: In fact,” be replaced by “. In fact,” or “: in fact,”
- 4) Line 635: briefly recall the formula or description of the mentioned statistical scores
Round 2
Reviewer 1 Report
REVIEW of the revised version of the manuscript "Satellite-based Flood Mapping through Bayesian Inference from a Sentinel-1 SAR Datacube" by Bernhard Bauer-Marschallinger, Senmao Cao, Mark Edwin Tupas, Florian Roth, Claudio Navacchi, Thomas Melzer, Vahid Freeman, Wolfgang Wagner [Remote Sens. 2022, 1,0. https://doi.org/].
The revised version of the manuscript is improved compared to its original version. The most of my comments are taken into account except that about the link between precipitation and flood. Because of that, this manuscript is now acceptable to be published in this journal after inclusion of the abovementioned material.
Author Response
Thank you again for the (rapid) feedback on our manuscript!
Concerning the link between precipitation and flood we are not completely sure how this is meant exactly. We feel that our addition in the last round around reference [6] points the interested reader to an excellent publication providing an overview on satellite precipitation observations used for riverine flood modelling, and we would prefer to not further extend the length of the introduction (this would open up a new line of narrative).
Reviewer 3 Report
As reference [48] is not yet published, this citation should be removed by the list of references and the text at lines 280-281 be modified accordingly.
Author Response
Thank you again for the (rapid) feedback on our manuscript!
We removed the reference that was before [48].